

# An improved similarity-based approach to predicting and mapping soil organic carbon and soil total nitrogen in a coastal region of northeastern China

Shuai Wang[1,2,3], Kabindra Adhikari[4], Qianlai Zhuang[3], Zijiao Yang[1], Xinxin Jin[1], Qiubing Wang[1] and Zhenxing Bian[1]

[1] College of Land and Environment, Shenyang Agricultural University, Shenyang, Liaoning, China
[2] Key Laboratory of Ecosystem Network Observation and Modeling, Institute of Geographic Sciences and Natural Resources Research, Chinese Academy of Sciences, Beijing, China
[3] Department of Earth, Atmospheric, and Planetary Sciences, Purdue University, West Lafayette, IN, United States of America
[4] Grassland, Soil and Water Research Laboratory, USDA-ARS, Temple, TX, United States of America

## ABSTRACT

Soil organic carbon (SOC) and soil total nitrogen (STN) are major soil indicators for soil quality and fertility. Accurate mapping SOC and STN in soils would help both managed and natural soils and ecosystem management. This study developed an improved similarity-based approach (ISA) to predicting and mapping topsoil (0–20 cm soil depth) SOC and STN in a coastal region of northeastern China. Six environmental variables including elevation, slope gradient, topographic wetness index, the mean annual temperature, the mean annual temperature, and normalized difference vegetation index were used as predictors. Soil survey data in 2012 was designed based on the clustering of the study area into six climatic vegetation landscape units. In each landscape unit, 20–25 sampling points were determined at different landform positions considering local climate, soil type, elevation and other environmental factors, and finally 126 sampling points were obtained. Soil sampling from the depth of 0–20 cm were used for model prediction and validation. The ISA model performance was compared with the geographically weighted regression (GWR), regression kriging (RK), boosted regression trees (BRT) considering mean absolute prediction error (MAE), root mean square error (RMSE), coefficient of determination ($R^2$), and maximum relative difference (RD) indices. We found that the ISA method performed best with the highest $R^2$ and lowest MAE, RMSE compared to GWR, RK, and BRT methods. The ISA method could explain 76% and 83% of the total SOC and STN variability, respectively, 12–40% higher than other models in the study area. Elevation had the largest influence on SOC and STN distribution. We conclude that the developed ISA model is robust and effective in mapping SOC and STN, particularly in the areas with complex vegetation-landscape when limited samples are available. The method needs to be tested for other regions in our future research.

Corresponding author
Xinxin Jin,
jinxinxin0218@syau.edu.cn

## INTRODUCTION

Soil organic carbon (SOC) and soil total nitrogen (STN) influence soil physical, chemical, biological properties and processes and determine soil quality that is related to food security by affecting agricultural productivity (*Batjes, 1996*; *Wang, Zhang & Li, 2013*; *Elbasiouny et al., 2014*). To a certain extent, SOC and STN affect the concentration of greenhouse gases in the atmosphere. The prediction of their sizes and changes has become a key climate change research area (*Jobbagy & Jackson, 2000*; *Post & Kwon, 2000*; *Lal, 2004*; *Davidson & Janssens, 2006*; *Yang et al., 2015a*; *Yang et al., 2015b*). Previous studies indicated that the spatially explicit information of SOC and STN plays an important role in quantifying global carbon and nitrogen cycles (*Kieft et al., 1998*; *Yang et al., 2015a*; *Yang et al., 2015b*; *Wang et al., 2017*).

Numerous studies have been conducted to understand the relationships between SOC and STN with environmental factors including topography, climate and biology (e.g., *Yang et al., 2016a*; *Yang et al., 2016b*). To map the spatial variations of SOC and STN based on field observations, Digital soil mapping (DSM) technologies have proven as a rapid and inexpensive approach over large areas using a limited amount of sample data (*McBratney, Santos & Minasny, 2003*; *Yang et al., 2016a*; *Yang et al., 2016b*; *Scull et al., 2003*; *Taghizadeh-Mehrjardi et al., 2014*). Commonly used DSM techniques include linear regression (*Kunkel et al., 2011*; *Zhao et al., 2015*), random forest (RF) (*Grimm et al., 2008*; *Yang et al., 2016a*; *Yang et al., 2016b*), regression kriging (RK) (*Odeh, McBratney & Chittleborough, 1995*; *Zhao et al., 2015*; *Song et al., 2016*), regression rules (*Adhikari et al., 2014*; *Minasny & McBratney, 2008*), boosted regression trees (BRT) (*Yang et al., 2016a*; *Yang et al., 2016b*; *Wang et al., 2016*), geographically weighted regression (GWR) (*Kumar, Lal & Liu, 2012*; *Song et al., 2016*; *Clement et al., 2009*), artificial neural networks (*Burke et al., 1989*; *Were et al., 2015*), similarity-based method (*Zhu, 1997*; *Yang et al., 2015a*; *Yang et al., 2015b*; *Liu et al., 2016*), and support vector machines (*Kovačević, Bajat & Gajić, 2010*; *Stevens et al., 2010*).

DSM conceptually applies the soil-landscape model of *Jenny (1941)* which describes the changes in soil properties as a function of factors related to climate, organism, topography, parent material and time. Therefore, the soil and its environment are specifically and spatially related. Topography plays an important role in the process of soil development and formation. Its impact on soil formation is mainly through the redistribution of water, nutrients, and energy, affecting microclimate and biomes, thus indirectly influencing the spatial distribution of SOC, and STN (*Adhikari et al., 2018*; *Mondal et al., 2017*; *Garten Jr & Ashwood, 2002*; *Moore et al., 1993*). In the central highlands of Ethiopia, *Tesfaye et al. (2016)* found topography as the main factor affecting the spatial variation of SOC and STN content in the region. Similarly, climatic influences on soil formation and development are mainly through temperature, and precipitation and their interactive effects. *Jobbagy & Jackson (2000)* applied three global-scale soil profile datasets to estimate the global SOC stocks in 3-m deep soil layer, and pointed out that temperature and precipitation were the main climatic factors affecting the changes in SOC stocks.

*Hudson (1992)* considered that soil-forming factors in a landscape interact with each other in a particular manner, indicating that the same soil-landscape presents with
homogeneous soil types. According to this theory, the relationship between soil and soil-forming factors can be used to infer the spatial distribution of soil properties. Similarly, *Zhu (1997)* suggested that soil-forming factors indicate the spatial distribution of soil properties. Thus, similar soil-environmental conditions can be assumed to exhibit similar soil types and properties. Based on this hypothesis, the complex environment can be divided into several less heterogeneous landscape units with potentially similar soil-forming environment. Soil properties in these landscape units can be further represented by the characteristics of the sample points within the landscape. The similarity-based method (*Zhu, 1997*) has been applied and proven to be an efficient method in predictive soil mapping. It has been widely used in the prediction of soil properties and soil types (*Liu et al., 2016*; *Liu, 2010*; *Zhu et al., 2010*). *Yang et al. (2015a)*; *Yang et al. (2015b)* used the similarity-based approach to map the spatial variation of soil salinity and obtained a higher prediction accuracy. A higher performance of similarity-based mapping approach was also confirmed by *Shi et al. (2004)*, who generated a soil series map with 86% prediction accuracy. *Liu (2010)* indicated that the similarity-based approach exhibited notable predictive performance compared to a kriging interpolation method applied in Ili of Xinjiang, China. Similarly, *Liu et al. (2016)* integrated a similarity-based approach with depth functions to model the three-dimensional (3D) distribution of SOM and obtained a lower global mean error (0.06 g kg$^{-1}$). Their findings showed that the similarity-based approach combined with other models can accurately predict soil properties.

The spatial distribution of SOC and STN in a landscape can be influenced by topography, organic matter input, temperature, humidity, vegetation, parent material and soil management  (*Sollins, Homann & Caldwell, 1996*; *Caminoserrano et al., 2014*), and can be better estimated by dividing the landscape into sub-units. To achieve this, some previous studies used data segmentation techniques to establish the relationship between SOC and STN with specific environmental variables for smaller regions. For instance, *Mulder, Lacoste & Richer-de Forges (2016)*  divided France into 10 soil-landscape types using a model-based clustering technology based on climate, vegetation, geology, and soil environmental variables and the regression models between SOC and environmental variables were established for each landscape. They found that the classification and prediction at each soil-landscape unit could better explain the SOC variations.

This study improved a similarity-based approach (ISA) to predict and map the spatial distribution of SOC and STN by applying soil property data and environmental variables. In the ISA model, the whole study area is clustered into several typical climatic-vegetation landscape units by using gaussian mixture model for model-based clustering (GMMC). The spatial prediction of SOC and STN was then carried out by using the similarity-based approach in this region. In the GMMC model, the expectation maximization algorithm is adopted to estimate the model parameters, whereas the Bayesian Information Criterion (BIC) is used to optimize the clustering results, the optimal model and the numbers of clusters are then selected. This study attempts to test the ISA model by comparing it with the commonly used DSM methods such as geographically weighted regression, regression kriging and boosted regression trees models. The proposed ISA model was tested to map SOC and STN in Montane ecosystems of Lushun City in the northeastern coastal areas of

China. Our research objectives to: (1) developed a GMMC method to divide the study area into climate-vegetation landscape units; (2) model the effect of environmental variables on SOC and STN variation; and (3) map the SOC and STN distribution using ISA, GWR, RK, and BRT models and compared their performances.

## MATERIALS & METHODS

### Study area

The study area (38.72°–38.97°N, 121.08°–121.47°E) covers approximately 512 km$^2$ and is located in Lushun City, Liaoning Province in the northeast coast of China (Fig. 1). The area lies in the southwest corner of Liaodong Peninsula and faces towards the Yellow Sea on the east and Bohai Sea on the west. Therefore, this region presents typical continental and oceanic climate characteristics forming unique climate-vegetation landscapes including coastal plain, interior plain, medium-coverage grasslands, high-coverage grasslands, low-elevation forest, and low-mountain shrub. The area is dominated by mountain landscape, which accounts for 53.1% of the total area and is characterized by valleys and peaks, with an altitude ranging from 137 to 466 m above sea level. The mean annual temperature (MAT) in this region is nearly 10 °C, the maximum and minimum temperatures are 27.5 °C in September and 8.2 °C in January. The region experiences a total of 185 frost-free days with the mean annual precipitation (MAP) ranging from 585 mm to 720 mm, 65–75% of which falls between June and September. According to World Reference Base for Soil Resources (WRB), the dominant soil types are Cambisols and Fluvisols (*Schad, Van Huyssteen & Micheli, 2014*).

### Dataset

#### Field sample data

Almost 53% of the study area is under forests, and is densely covered with rivers and valleys where the river system is rather complicated in a rugged terrain. To represent the spatial characteristics of soil properties in such a complex landscape, we applied a stratified sampling scheme following two steps.

First, we used a GMCC method to divide the study area into several typical climatic-vegetation landscape units. The GMMC model is a hybrid model composed of the gaussian mixture model (GMM) and the expectation maximization (EM) algorithm (*Dempster, Laird & Rubin, 1977*). The GMM model is used to determine certain probability density functions (i.e., the probability of data points is partitioned into each category) (*Banfield & Banfield, 1993*) to achieve the division of datasets. The EM algorithm is adopted to estimate the model parameters, whereas the BIC is used to optimize the clustering results, and then the optimal model and the number of clusters are selected. The clustering model is established in the R version 3.2.2 (*R Development Core Team, 2013*) using the ''mclust'' package (*Fraley & Raftery, 2002*) for completion. Based on the pedogenetic information of the study area, environmental factors such as elevation, MAT, MAP, and normalized difference vegetation index (NDVI) were used as inputs (*Zhu et al., 2008*; *Yang et al., 2013*; *Yang et al., 2016a*; *Yang et al., 2016b*). The study area was divided into six climate-vegetation landscape units (Fig. 2). Within each climate-vegetation landscape, twenty to twenty-five
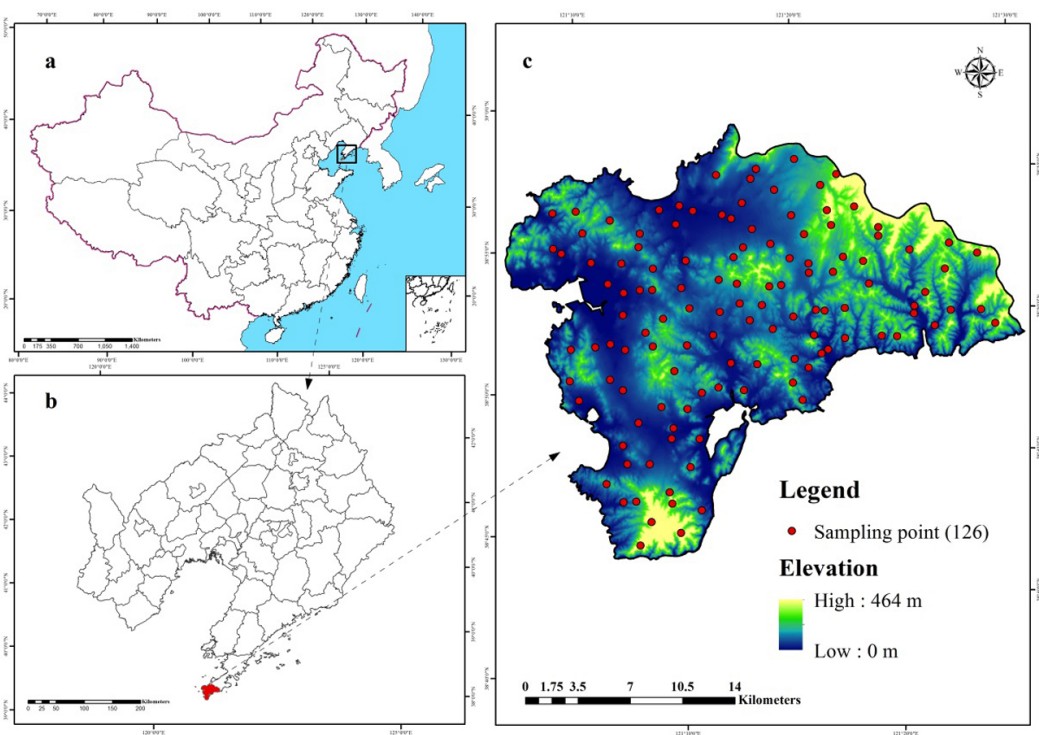

**Figure 1** Location of the study area and 126 sampling sites (C) in Liaoning Province (B), China (A), which are shown superimposed on a 30-m resolution digital elevation model.

sampling points were identified at different landform positions considering local climate, soil type, elevation and other environmental factors.

Although soil sampling in this study considered climate and vegetation properties of the landscape for clustering, it was further verified by the local soil experts to determine whether the units are typical and are representative of the study area. This data sampling was not based on probability sampling strategy, thus it might underestimate or overestimate the SOC and STN distribution in the region as reported in previous studies (*Zhu, 1997*; *Yang et al., 2016a*; *Yang et al., 2016b*; *An et al., 2018*). However, it was probably the best sampling option given that sampling was done in such a densely forested terrain and with limited resources. Overall flowchart of the proposed methodology is shown in Fig. 3.

A total of 126 sample locations were established, and the geographic coordinates of each point was recorded by a handheld global positioning system (GPS) (Table S1). From each sample location, about 1 kg of topsoil (0–20 cm soil depth) sample was collected for laboratory analysis. In the Testing & Analysis Center of Shenyang agricultural University, Shenyang, Liaoning Provence, China, the litters were removed from samples. The samples were then air dried, grinded, and passed through a 2-mm sieve. SOC and STN contents were measured by a dry combustion method (*Matejovic, 1993*) using CN analyzer (Vario Max, Elementar Amerivas Ins., Germany). The soil depth considered in this study was limited to 0–20 cm because *Wang et al. (2017)* found that in Liaoning Province, 69% of SOC and STN were stored in the topsoil (0–30 cm). It has been further confirmed by Liu

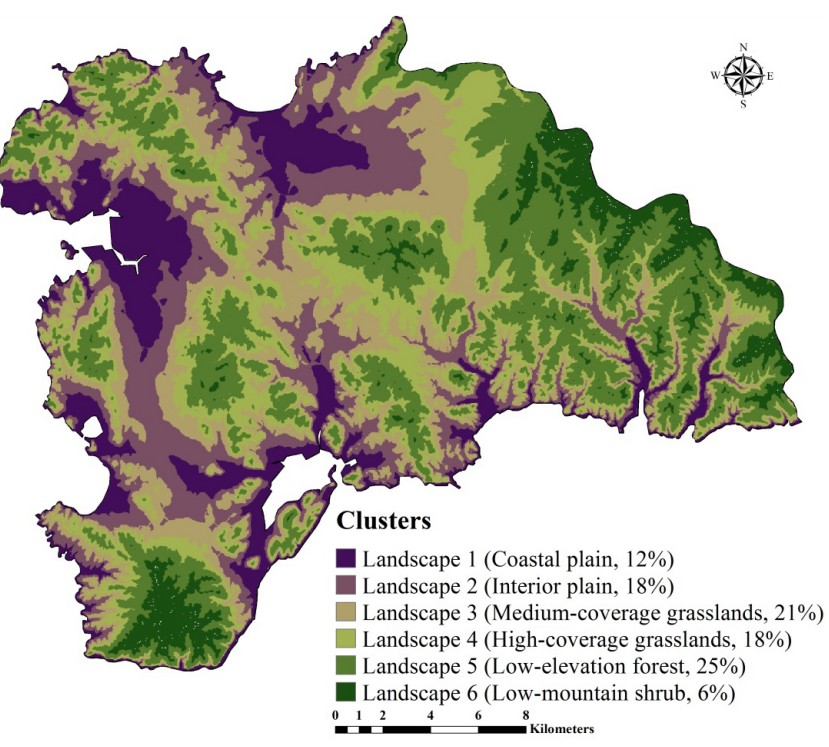

**Figure 2** Climate-vegetation landscapes in the study area: Landscape 1, coastal plain; Landscape 2, interior plain; Landscape 3, medium-coverage grasslands; Landscape 4, high-coverage grasslands; Landscape 5, low-elevation forest; Landscape 6, low-mountain shrub.

et al. (2012) who reported about 43% of the SOC stocks from the topsoil. In addition, 70% of all samples were randomly selected as training dataset ($n = 88$) and the rest 30% as independent verification dataset ($n = 38$).

### Environmental variables

Two topographic variables including slope gradient and topographic wetness index (TWI) were used in addition to four other variables (elevation, MAP, MAT, and NDVI) for climate-vegetation landscape partition (Table 1), to predict the spatial distribution of SOC and STN content. The variables were collected from different sources and were converted to a raster grid of 30 m resolution. Measured data on SOC and STN, and all the predictors were brought into the geographic information system (GIS) in a common projection system (Krasovsky_1940_Albers) in ArcGIS 10.2 (ESRI Inc., USA) for further geospatial processing and analysis. Slope gradient and TWI were derived from a 30-m resolution digital elevation model (DEM) captured from Shuttle Radar Topography Mission (SRTM) (*Farr & Kobrick, 2000*; *Conrad et al., 2015*). MAT and MAP as two main climatic variables obtained from China Meteorological Data Service Center (http://data.cma.cn/en) as 30-year annual average (1980–2010). In addition, we selected NDVI to represent vegetation intensity, as determined by using two bands of Landsat 5, namely, band 3 and band 4 (Eq. (1)), and were downloaded from the United States Geological Survey (USGS). The imagery covers from

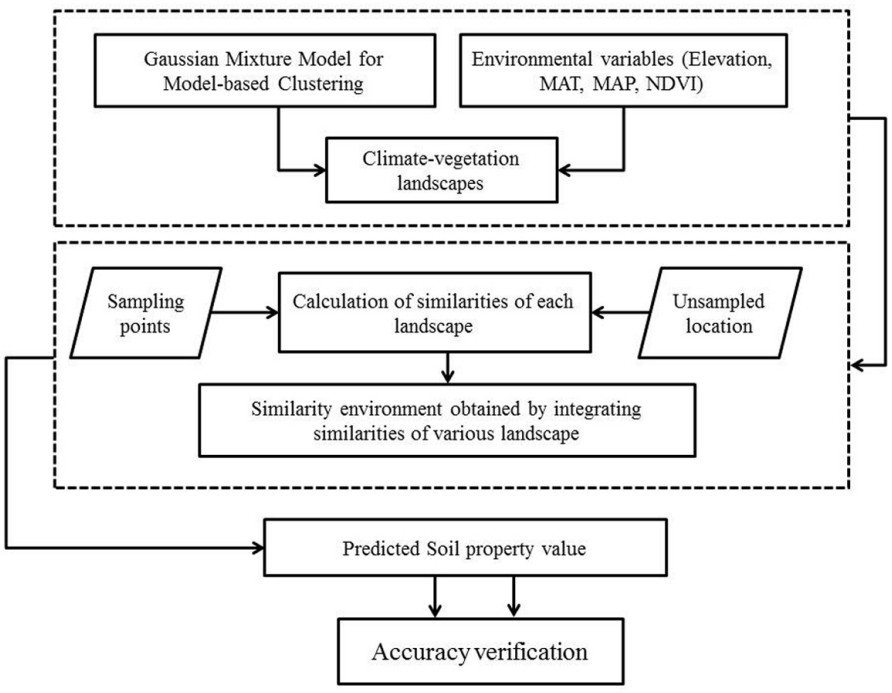

**Figure 3** **Flowchart of the methodology adopted in the study.**

July to September of 2012 with <10% cloud cover. Image processing included geometric correction, registration, image mosaic and clipping, cloud removal, and shadow processing and was performed in ENVI 4.7 software (*Huang et al., 2007*).

$$NDVI = \frac{(band4 - band3)}{(band4 + band3)} \tag{1}$$

## Model development
### *Improved similarity-based approach and its prediction consistency*
An improved similarity-based approach was used to estimate the soil properties of unsampled location by acquiring the environmental similarity with sampling points. Specifically, this method was developed following three key steps:

Step 1 was to calculated the environmental similarity in each climate-vegetation landscape;

$$D = \sqrt{(S_1 - U_1)^2 + (S_2 - U_2)^2 + \ldots + (S_i - U_i)^2} \tag{2}$$

In this equation, $D$ is Euclidean distance; $Si$ and $Ui$ are sampled point $S$ and unsampled point $U$, respectively, and the variable values in dimension $i$ space. Suppose there are $m$ sampling points and $n$ unsampled points, then a set of Euclidean distance ($DE$) matrix can

**Table 1  Summary statistics of SOC, STN, and environmental variables at sampling points.**

| Property | Description | Units | Minimum | Mean | Maximum | SD | Skewness | Kurtosis |
|---|---|---|---|---|---|---|---|---|
| SOC | Level of organic carbon content in soil | g kg$^{-1}$ | 8.50 | 12.95 | 22.02 | 3.78 | 1.07 | 0.06 |
| STN | Sum of various forms of nitrogen in soil | g kg$^{-1}$ | 0.57 | 0.85 | 1.49 | 0.23 | 0.81 | −0.31 |
| NDVI | The difference between near infrared and red band reflectance is divided by the sum of their | index | 0.15 | 0.40 | 0.56 | 0.08 | −0.25 | 0.15 |
| MAT | Mean annual temperature from 1980 to 2010 | Celsius degree | 9.5 | 10.39 | 10.91 | 0.26 | −1.29 | 1.80 |
| MAP | Mean annual precipitation from 1980 to 2010 | mm | 605.8 | 609.2 | 622.6 | 3.97 | 1.54 | 1.95 |
| Elevation | Absolute vertical distance to geoid | m | 1 | 66.53 | 309 | 61.48 | 1.67 | 2.87 |
| Slope gradient | Maximum rate of change between cells and neighbors | degree | 0 | 9.46 | 41.90 | 9.78 | 1.33 | 1.29 |
| TWI | Calculates slope and specific catchment area based topographic wetness index | index | 2.33 | 5.04 | 10.13 | 1.87 | 0.89 | 0.52 |

**Notes.**
SOC, soil organic carbon; STN, soil total nitrogen; TWI, topographic wetness index; MAP, mean annual precipitation; MAT, mean annual temperature; NDVI, Normalized Difference Vegetation Index; SD, Standard deviation.

be obtained as:

$$DE = \begin{bmatrix} D_{11}, & D_{21}, & \ldots, & D_{n1} \\ D_{12}, & D_{22} & \ldots, & D_{n2} \\ \ldots \\ D_{1m} & D_{2m} & \ldots, & D_{nm} \end{bmatrix} \tag{3}$$

Euclidean distance only measures the degree of distance between two points. In order to obtain the similarity between two points, it must be converted into the corresponding similarity (*Danielsson, 1980*). We first applied Eq. (4) to obtain standard range of Euclidean distance ($DE_{scale}$) [0,1], and then applied Eq. (5) to convert it into similarity ($SE$) with range [0, 1].

$$DE_{scale} = DE / \max(DE) \tag{4}$$

$$SE = 1 - DE_{scale} \tag{5}$$

So, the similarity matrix (*SE*) between the unsampled locations and the sampling points was obtained using Eq. (6):

$$SE = \begin{bmatrix} S_{11}, & S_{21}, & \ldots, & S_{n1} \\ S_{12,} & S_{22} & \ldots, & S_{n2} \\ \ldots & & & \\ S_{1m} & S_{2m} & \ldots, & S_{nm} \end{bmatrix} \tag{6}$$

Step 2 was to predict soil properties for the landscape unit according to its environmental similarity.

In each landscape, the soil property value of the unsampled locations was predicted by using the environmental similarity where elevation, MAT, MAP, NDVI, TWI, and slope gradient were used as predictors. The definite equation as follows:

$$V_b = \sum_{a=1}^{m} S_{ab} \cdot V_a / \sum_{a=1}^{m} S_{ab} \tag{7}$$

where $V_b$ is the predicted value of soil property at location $b$ and $V_a$ is the soil property value at sampled location $a$; $S_{ab}$ is the environmental similarity between location $b$ and sampled location $a$; $m$ is the number of sampling points.

Step 3 was to calculate the prediction consistency using Eq. (8):

$$Consistency_m = 1 - \max(S_{1m}, S_{2m}, \ldots, S_{nm}) \tag{8}$$

where $Consistency_m$ is the prediction consistency at point $m$, and the lower the value, the better the prediction consistency of the model; $S_{nm}$ is the environmental similarity between point $n$ and sampled location $m$.

According to Eq. (8), the maximum value can be used to obtain the best representation of the similarity of the environment between unsampled locations and sampling points (i.e., most similar). With a low similarity, the sampling points cannot represent the unsampled locations (*Liu, 2010*) and the soil property values inferred from the sampling points will show a high degree of speculative consistency.

### Geographically weighted regression

Geographically weighted regression was first introduced into the study of geography by *Brunsdon, Fotheringham & Chariton (1998)*. It embeds the spatial structure of the data into a regression model making the regression parameters become a function of the geographical location of the observation points. GWR is a non-parametric technique based on local weighted regression developed for curve fitting and smoothing applications in statistics (*Kumar, Lal & Liu, 2012*). In this technique, local regression parameters are estimated using a subset of data close to the estimated points of models in variable spaces. The innovation of GWR is to use a subset of data in geographic space near the calibration location of the model. In our GWR model, six environmental variables were used to predict the spatial distribution of SOC and STN.

### Regression kriging

Regression kriging is a spatial interpolation technique that combines a regression of the dependent variable and auxiliary variables (such as terrain parameters, remote sensing

imagery and thematic maps) with kriging of the regression residuals (*Odeh, McBratney & Chittleborough, 1995*; *Hengl, Heuvelink & Stein, 2004*). We applied RK to interpolate the spatial distribution of SOC and STN following these five steps: (1) determining the SOC and STN prediction model using multiple linear regressions (MLR); (2) calculating the SOC and STN prediction model residuals at each calibration location; (3) modeling the covariance structure of the SOC and STN residuals using a variogram GS+ 7.0 statistical software (Gamma Design Software, Plainwell, MI) was used to implement this process; (4) spatially interpolating the SOC and STN residuals through the parameters of the variogram model; and (5) adding the SOC and STN prediction model surface to the interpolated residuals surface to get the final predicted map.

### *Boosted regression trees*

Boosted regression trees model is a machine learning algorithm based on classification and regression trees, which was proposed by *Friedman, Hastie & Tibshirani (2000)*. The BRT model is similar to other boosting models improving model performance by training multiple models and combining them for prediction. It consists of two algorithms: regression trees and gradient boosting. In the model, gradient lifting algorithm is used to linearly combine multiple regression trees of weak regression to form an efficient and a strong regression model (e.g., *Wang et al., 2016*; *Yang et al., 2016a*; *Yang et al., 2016b*). The implementation of BRT model requires users to define four parameters: learning rate (LR), tree complexity (TC), bag fraction (BF) and tree number (NT). LR represents the contribution of each tree in the model to the final fitting model (*Yang et al., 2016a*; *Yang et al., 2016b*). TC is a direct predictor of tree depth and maximum interaction level (*Yang et al., 2015a*; *Yang et al., 2016b*). BF represents the scale of data used in each model (*Wang et al., 2018a*; *Wang et al., 2018b*). NT is determined by LR and TC. In order to obtain the best prediction performance of BRT model, different parameter combinations LR (0.0025, 0.025, 0.25, 0.50), TC (3, 6, 9), BF (0.20, 0.35, 0.50, 0.65), NT (600, 800, 1000, 1200) were tested by 10-fold cross-validation. Finally, LR, TC, BF and NT values that achieved the minimum prediction error through 10-fold cross-validation were set to 0.0025, 6, 0.65, and 800, respectively, for SOC prediction, and 0.025, 6, 0.65 and 1000, respectively, for STN prediction.

## Model validation

In each landscape, 30% sampling points (38 observations in total) were randomly selected to test the prediction performance of the ISA, GWR, RK, and BRT methods. Three commonly-used indices including mean absolute prediction error (MAE), root mean square error (RMSE), coefficient of determination ($R^2$), and maximum relative difference (RD) (Eqs. (9) to (11)) were used to compare the model. All indices were calculated in the R version 3.2.2 (*R Development Core Team, 2013*):

$$MAE = \frac{1}{n} \sum_{i=1}^{n} |a_i - b_i| \qquad (9)$$

$$RMSE = \sqrt{\frac{1}{n} \sum_{i=1}^{n} (a_i - b_i)^2} \qquad (10)$$

$$R^2 = \frac{\sum_{i=1}^{n} \left(a_i - \overline{b}_i\right)^2}{\sum_{i=1}^{n} (b_i - b_i)^2} \qquad (11)$$

$$RD = \frac{\overline{a}_i - \overline{b}_i}{\overline{a}_i} \qquad (12)$$

where $a_i$, $b_i$, $\overline{a}_i$, and $\overline{b}_i$ are the observed values, predicted values, and mean values of soil property at site $i$, and $n$ is the number of the sampling point.

## RESULTS

### Exploratory Data Analysis

Exploratory analysis of measured SOC and STN contents and values of environmental variables at sampling locations are summarized in Table 1. Average SOC and STN in the topsoil were ($\pm 3.78$), and 0.85 ($\pm 0.23$) g kg$^{-1}$, respectively. The skewness and kurtosis coefficients were 1.07 and 0.06 for SOC and 0.81 and -0.31 for STN, respectively, indicating that the measured SOC and STN approximately followed a normal distribution. In addition, both distribution passed Kolmogorov–Smirnov (K-S) test ($\rho = 0.11$ and 0.09), respectively, suggesting that SOC and STN data did not need to be transformed for subsequent analysis and modeling.

SOC and STN were positively correlated with NDVI, MAP, elevation, and slope gradient but negatively correlated with MAT and TWI (Fig. 4). We observed multicollinearity among environmental variables and believed that predicting SOC and STN with these variables using traditional statistical methods such as simple multiple regression equations would be unreliable (*McBratney, Santos & Minasny, 2003*; *Yang et al., 2016a*; *Yang et al., 2016b*) and using ISA model could effectively overcome this problem. In the process of GWR modeling, we eliminated the slope gradient variable as it was highly correlated with MAT and elevation, to avoid the problem of multicollinearity, and used the remaining five variables for spatial prediction.

### *Climate-vegetation landscape*

SOC and STN usually vary in the long term and are influenced by various environmental factors (i.e., rainfall, temperature, and vegetation). As a typical coastal ecosystem in Northeast China, Lushun has both continental and marine climate characteristics with an absence of cold winter and hot summer. Therefore, we selected the elevation, MAT, MAP, and NDVI to divide the region into the climate-vegetation landscape units. The results showed that the study area can be divided into six climate-vegetation landscapes (BIC, -678247.3) (Fig. 2). Landscapes 1, 2 are mainly distributed in the northern and

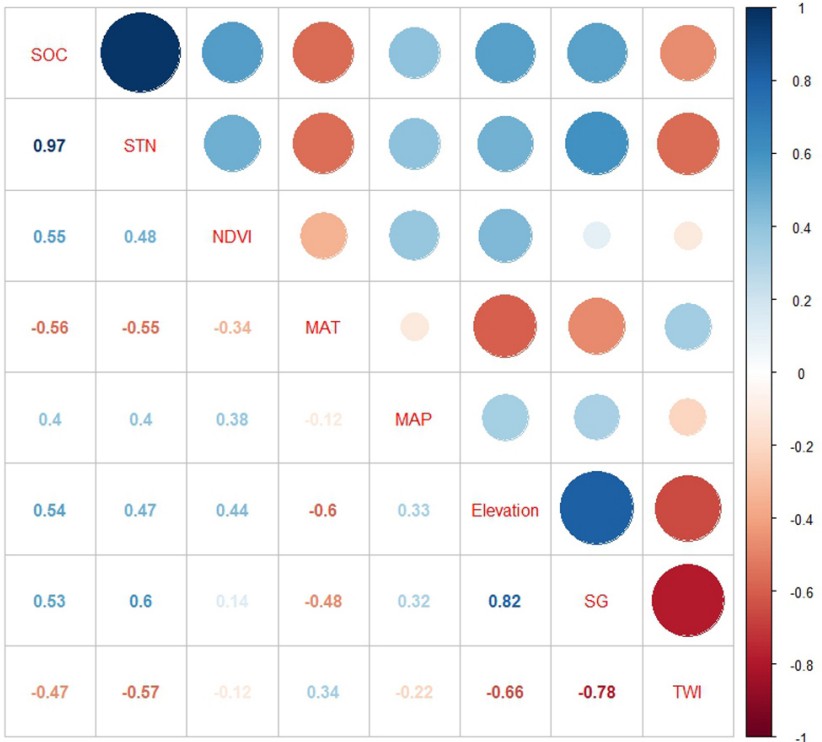

**Figure 4** **Pearson correlation coefficients of SOC and STN with environmental variables: SOC, soil organic carbon; STN, soil total nitrogen; TWI, topographic wetness index; SG, slope gradient; MAP, mean annual precipitation; MAT, mean annual temperature; and NDVI: Normalized Difference Vegetation Index.**

western region of the study area; Landscapes 3 and 4 are mainly for the central region; and Landscapes 5, 6 are mainly distributed in the northeastern and southern regions. In terms of the area distribution area of landscapes, landscape 5 accounting for 25% of the total area of the study area, followed by landscape 3 (21%), landscape 2 (18%), landscape 4 (18%), landscape 1 (12%), and landscape 6 (6%) (Fig. 2).

The distribution of climate-vegetation landscapes and soil observations by elevation, MAP, MAT, and NDVI is shown in Fig. 5. The statistical characteristics of the four environmental variables at sampling points were similar to those in the landscapes, indicating that the sampling points could efficiently describe the characteristics of the major environmental factors in the study area. However, some differences in the main environmental variables of each landscape were observed. The Landscape 1 showed the lowest average elevation, a low MAP (608 mm), and the highest MAT (10.6 °C), and the average NDVI value approached 0.4. Therefore, the landscape type was coastal plain. Compared with landscape 1, landscape 2 was distributed at a relatively low elevation but presented a better vegetation cover; therefore, the landscape type was named interior plain. Landscape 6 showed the highest elevation (210 m), lowest MAT (10.0 °C), and lowest value of NDVI; thus, the landscape type was low-mountain shrub. Landscapes 3, 4, and 5 were

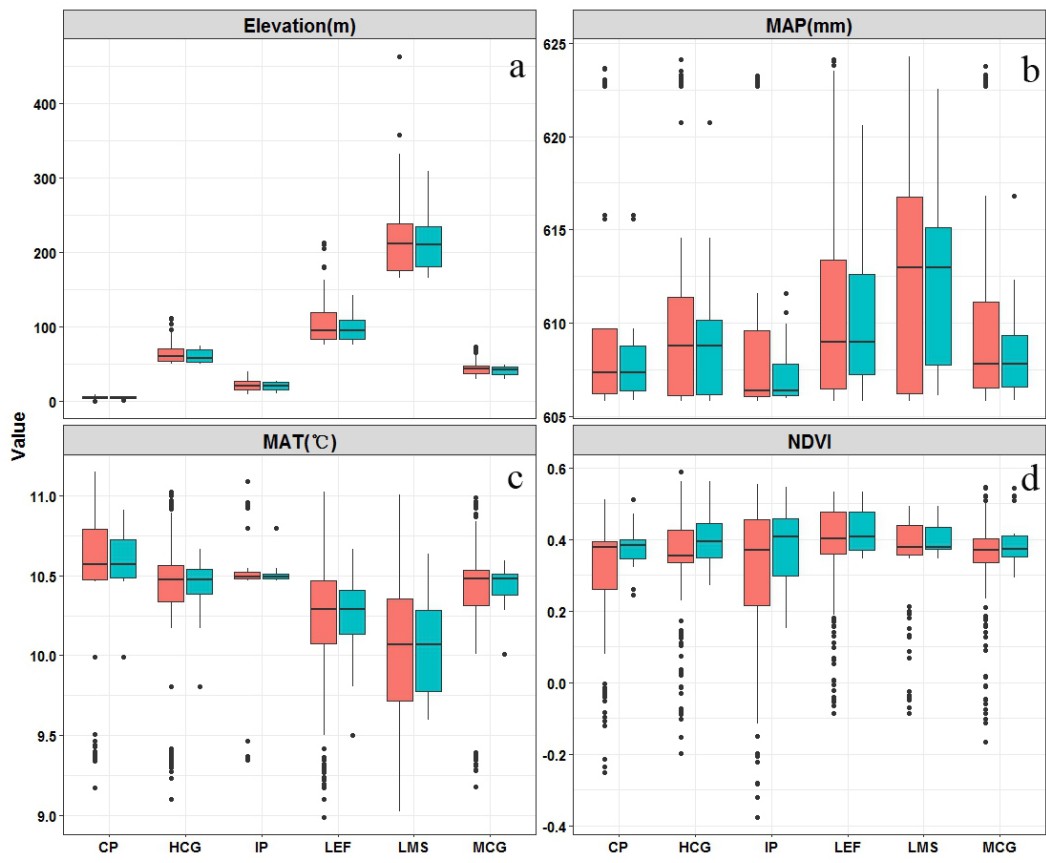

**Figure 5** Boxplots of main environmental features related (elevation (A), mean annual precipitation (MAP) (B), mean annual temperature (MAT) (C), and normalized difference vegetation index (NDVI)) to climate-vegetation landscapes including CP: coastal plain; IP: interior plain; MCG: medium-coverage grasslands; HCG: high-coverage grasslands; LEF: low-elevation forest; and LMS: low-mountain shrub. Red is the training dataset ($n = 88$) and green is the independent verification dataset ($n = 38$).

mainly distributed in the middle elevation, but their MAT and NDVI values were different. Landscapes 3 and 4 showed higher precipitation rates; however, given the limitation of temperature, low vegetation cover was observed, thus landscapes 3 and 4 were classified as medium-coverage grasslands and high-coverage grasslands. Landscape 5 presented a better vegetation cover, and was classified as low-elevation forest.

## Model performance

In order to determine whether the study area should be divided into unique landscape units, and then use the best prediction model to predict the spatial distribution of SOC and STN in each landscape, the MAE, RMSE, and $R^2$ were compared between the observed and predicted values (Table 2). The results showed that the model was significantly improved by dividing the study area into climate-vegetation landscapes (Table 2).

To obtain the best prediction model, four models (ISA, GWR, RK, and BRT) were compared using 38 independent validation points. Results showed that the ISA model

**Table 2 Comparison of the performances of ISA, GWR, RK, and BRT models using MAE, RMSE, and $R^2$ with validation Data in predicting SOC and STN.**

| Property | Category | Item | MAE | RMSE | $R^2$ |
|---|---|---|---|---|---|
| SOC (g kg $^{-1}$) | Classified | ISA | 0.74 | 1.01 | 0.76 |
| | | GWR | 0.96 | 2.35 | 0.57 |
| | | RK | 1.12 | 2.87 | 0.48 |
| | | BRT | 0.87 | 1.68 | 0.64 |
| | Unclassified | ISA | 0.98 | 1.34 | 0.54 |
| | | GWR | 1.17 | 2.67 | 0.46 |
| | | RK | 1.27 | 2.97 | 0.41 |
| | | BRT | 1.05 | 2.13 | 0.51 |
| STN (g kg $^{-1}$) | Classified | ISA | 0.03 | 0.04 | 0.83 |
| | | GWR | 0.08 | 0.09 | 0.54 |
| | | RK | 0.11 | 0.15 | 0.43 |
| | | BRT | 0.06 | 0.07 | 0.67 |
| | Unclassified | ISA | 0.12 | 0.09 | 0.57 |
| | | GWR | 0.13 | 0.16 | 0.41 |
| | | RK | 0.15 | 0.19 | 0.38 |
| | | BRT | 0.11 | 0.13 | 0.51 |

**Notes.**

SOC, soil organic carbon; STN, soil total nitrogen; MAE, mean absolute prediction error; RMSE, root mean square error; $R^2$, coefficient of determination; GWR, geographically weighted regression; RK, regression kriging (RK); BRT, boosted regression trees.

outperformed GWR, RK, and BRT models with a lower MAE and RMSE, and a higher $R^2$ values. We also found that the performance of BRT was better than the GWR and RK models. ISA model was efficient and powerful in spatial prediction of SOC, and STN among six climatic vegetation landscape units based on the accuracy verification. The prediction accuracy of ISA model in each landscape is listed in Table 3. The mean predicted values of SOC and STN in the topsoil were comparable to the mean observed values of the sampling points (Table 3). RD was obtained in the interior plain (RD about 7.1% and 4.2%), followed by the coastal plain (RD about 3.4% and 3.3%) and low-mountain shrub (RD about −4.2% and −4.1%), and the RD values of the other landscapes were smaller (Table 3). The predicted $R^2$ ranged from 0.37 (Low-mountain shrub) to 0.77 (Interior plain) for SOC and 0.39 (Low-mountain shrub) to 0.89 (Median-coverage grassland) for STN, respectively. The lowest RMSE (0.13 g kg$^{-1}$) in SOC prediction was reported for Medium-coverage grassland and the highest (2.53 g kg$^{-1}$) for Low-elevation forest. For the STN, RMSE ranged from 0.01 to 0.06 depending on the landscape types. Overall, the lower MAE and RMSE, and higher $R^2$ value for ISA model compared to the rest of the models indicated that the ISA model could better predict SOC and STN distribution.

For SOC prediction using the ISA model, MAE, RMSE, and $R^2$ values were 0.74, 1.01 g kg$^{-1}$, and 0.76, respectively, and for STN, the values were 0.03, 0.04 g kg$^{-1}$, and 0.83, respectively (Fig. 6). The ISA model overestimated the SOC and STN contents in the study area, but the overall accuracy was high. The model explained approximately 76% and 83% of the total SOC and STN variability in the study area, respectively. Predictive
**Table 3  Statistics of prediction performance and mean prediction results of each climate-vegetation landscapes by ISA model.**

| Property | Index | Unit | Climate-vegetation landscapes | | | | | |
|---|---|---|---|---|---|---|---|---|
| | | | Coastal plain | Interior plain | Medium-coverage grasslands | High-coverage grasslands | Low-elevation forest | Low-mountain shrub |
| SOC | Observed | (g kg$^{-1}$) | 9.11 | 11.04 | 11.29 | 12.38 | 14.41 | 20.02 |
| | Predicted | (g kg$^{-1}$) | 8.80 | 10.26 | 11.06 | 12.19 | 14.78 | 20.86 |
| | RD | (%) | 3.40 | 7.07 | 2.04 | 1.53 | −2.57 | −4.20 |
| | MAE | (g kg$^{-1}$) | 0.31 | 0.81 | 0.28 | 0.58 | 1.44 | 0.84 |
| | RMSE | (g kg-1) | 0.14 | 1.56 | 0.13 | 0.46 | 2.53 | 0.82 |
| | $R^2$ | | 0.64 | 0.77 | 0.6 | 0.55 | 0.49 | 0.37 |
| STN | Observed | (g kg$^{-1}$) | 0.61 | 0.72 | 0.76 | 0.83 | 0.99 | 1.22 |
| | Predicted | (g kg$^{-1}$) | 0.59 | 0.69 | 0.76 | 0.81 | 1 | 1.27 |
| | RD | (%) | 3.28 | 4.17 | 0.00 | 2.41 | −1.01 | −4.10 |
| | MAE | (g kg$^{-1}$) | 0.03 | 0.04 | 0.01 | 0.03 | 0.03 | 0.05 |
| | RMSE | (g kg$^{-1}$) | 0.01 | 0.04 | 0.01 | 0.02 | 0.06 | 0.02 |
| | $R^2$ | | 0.76 | 0.82 | 0.89 | 0.5 | 0.89 | 0.39 |

**Notes.**

SOC, soil organic carbon; STN, soil total nitrogen; RD, relative difference; MAE, mean absolute prediction error; RMSE, root mean square error; $R^2$, coefficient of determination.

consistency maps of SOC and STN were obtained using Eq. (8). The higher the consistency value, the larger the difference between the predicted value and measured value. Mean prediction consistency values were 0.14 for SOC and 0.13 for STN, respectively (Figs. 7A, 7B), suggesting the ISA model had a reliable predictability.

### Estimates of SOC and STN

SOC and STN distribution was mapped using the ISA model where an average SOC and STN content in the study area was 11.37 (±3) and 0.89 (±0.2) g kg$^{-1}$, respectively (Figs. 7C, 7D). In terms of spatial pattern, ISA model was more excellent than other models (Fig. S1). Among the climate-vegetation landscape units, soils under low-elevation forests contained more SOC and STN than the rest of the landscape units (Table 4). Of all predictors, elevation was the main predictor affecting the spatial distribution of SOC and STN in the study area (Fig. 8).

## DISCUSSION

### Effects of ISA model on SOC and STN

The study area divided into unique landscape units, and then used the best prediction model to predict the spatial distribution of SOC and STN in each landscape. The results showed that the model was significantly improved by dividing the study area into climate-vegetation landscapes. This is consistent with previous studies using the same method. For instance, in Belgium, *Lettens et al. (2004)* divided the whole study area into 289 landscape units and predicted the SOC stocks for each landscape unit. They considered that SOC stocks were continuously influenced by a number of external factors, mainly land-use history and current land management and climate. The spatial distribution of SOC and STN varies closely with climate-vegetation-dominated landscapes. By dividing a large-scale complex

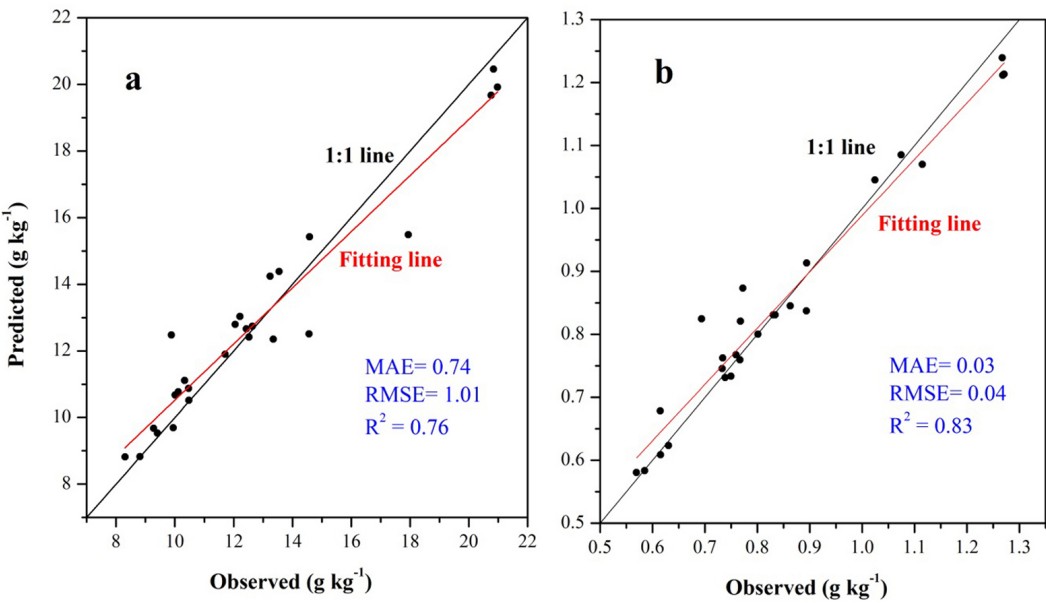

**Figure 6** Scatter plot between the observed soil organic carbon (SOC) (A) and soil total nitrogen (STN) (B) content with its predicted values using the ISA model based on independent validation samples.

landscape into small units, the SOC and STN could be better predicted. At the same time, compared to other SOC and STN prediction studies in the coastal area, the ISA model we constructed performed better. For example, in Santa Cruz Island, Galapagos, *Rial et al. (2017)* used terrain, climate and remote sensing variables combined with a GWR method to predict topsoil SOC stocks, and their model could explain 66% of SOC stocks variation in the region. In a separate study, *Wang et al. (2018b)* compared GWR and RK models to predict topsoil SOC in the Northeast coastal area of China and found GWR a better model that could explain 78% and 80% SOC variation during 1982 and 2012, respectively. The ISA model could explain approximately 76% and 83% of the total SOC and STN variability in the study area, respectively. In comparison with previous studies, we observed a better prediction performance of the ISA method. *Yang et al. (2016a)* and *Yang et al. (2016b)* developed a BRT model that explained 50 to 58% of the SOC variability in the Qinghai Tibet Plateau, China. Using a GWR approach, *Wang, Zhang & Li (2013)* captured approximately 57% of the STN variability in Fujian Province, China. In the Medinipur Block, Paschim Medinipur District and West Bengal in India, *Bhunia, Shi & Pourghasemi (2019)* used remote sensing techniques and multivariate regression model, which explained 71% of SOC variability. *Xu et al. (2018)* used six different remote sensing spectral indices of RK models to predict STN in two smallholder villages, Kothapally and Masuti in South India, which helps explain 59% of the STN variability in the region.

Overall, the ISA model prediction consistency was higher towards the eastern part of the study area compared to the rest area for both SOC and STN predictions. Although there were other sources of uncertainty and error in mapping SOC and STN, including sampling

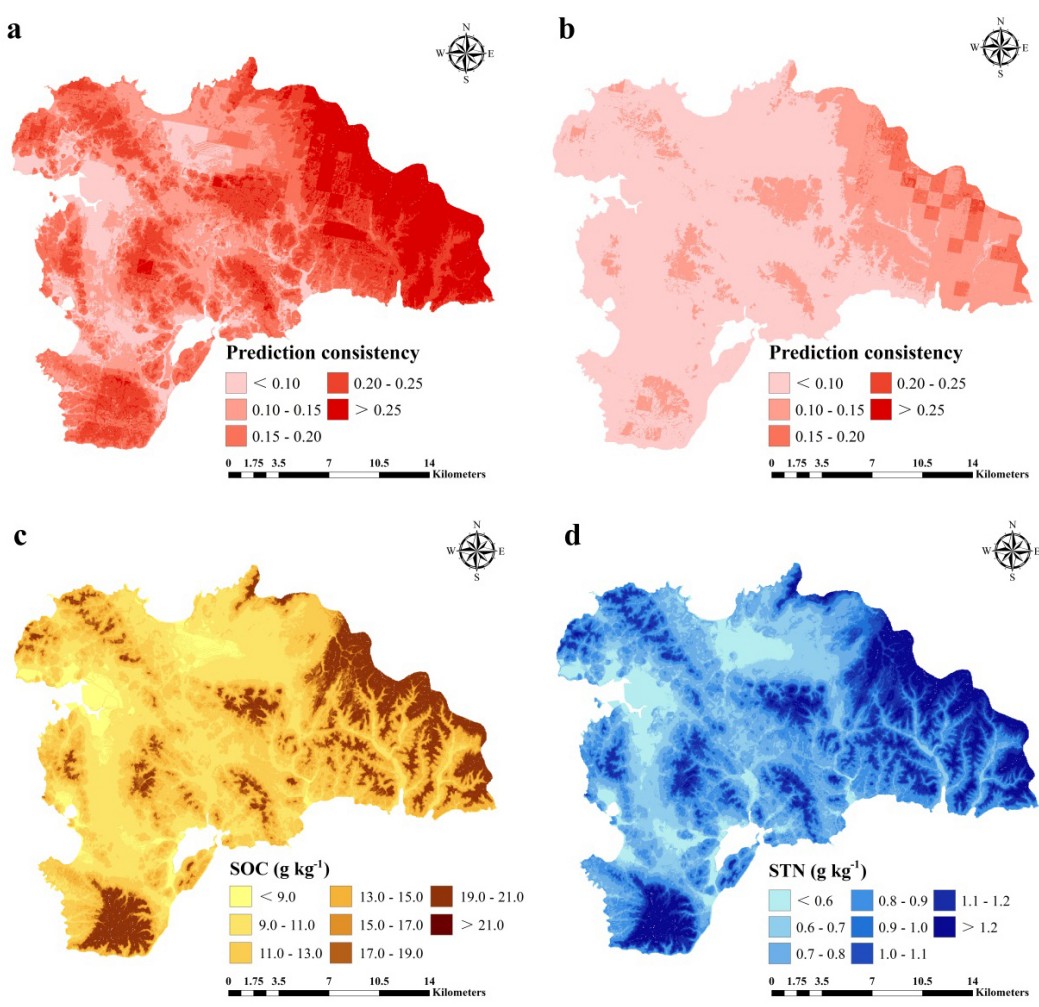

**Figure 7** Prediction consistency map of SOC (A) and STN (B) and spatial distribution of SOC (g kg$^{-1}$) (A) and STN (g kg$^{-1}$) (B) predicted using an improved similarity-based approach (ISA).

error, laboratory analysis errors, and low predictor precision (*Wang et al., 2017*; *McBratney, Santos & Minasny, 2003*; *Yang et al., 2016a*; *Yang et al., 2016b*; *Grimm et al., 2008*; *Fuchset et al., 2009*), quantification of these inevitable errors were not considered in this study.

## Predicted distribution of SOC and STN content

In general, the spatial distribution pattern for SOC and STN was similar and varies with different climate-vegetation landscape units (Figs. 3, 5 and 7). Spatially, higher SOC and STN were in the northeastern and southern regions that were dominated by low-elevation forest and low-mountain shrub. Coastal plain has the lowest SOC and STN. Overall, the lowest SOC and STN contents were found in the northern and central regions with low MAT and vegetation coverage. These results are consistent with the findings of *Jobbagy & Jackson (2000)* and *Ding et al. (2018)* who reported for humid and rainy areas. Soil particles in high-elevation areas were easy to lose and thick soil layers were formed in

**Table 4** Summary statistics of soil organic carbon (SOC) and soil total nitrogen (STN) prediction in different climate-vegetation landscapes using the ISA method based on 38 independent validation points.

| Climate-vegetation landscapes | Number of independent validation points | SOC (g kg⁻¹) | | STN (g kg⁻¹) | |
|---|---|---|---|---|---|
| | | Range of change | Mean ± SD | Range of change | Mean ± SD |
| Coastal plain | 5 | 0.85–17.64 | 6.97 ± 5.47 | 0.34–1.13 | 0.63 ± 0.62 |
| Interior plain | 7 | 2.38–33.98 | 9.68 ± 6.69 | 0.43-1.32 | 0.8 ± 0.69 |
| Medium-coverage grasslands | 6 | 7.61–29.74 | 11.12 ± 6.32 | 0.74–2.07 | 0.88 ± 0.67 |
| High-coverage grasslands | 7 | 8.04–33.25 | 12.14 ± 6.91 | 0.77–2.28 | 0.94 ± 0.7 |
| Low-elevation forest | 8 | 10.24–33.8 | 16.84 ± 6.14 | 0.9–2.31 | 1.22 ± 0.66 |
| Low-mountain shrub | 5 | 7.32–26.33 | 11.49 ± 5.34 | 0.73–1.86 | 0.91 ± 0.61 |

**Notes.**
SD, Standard Deviation.

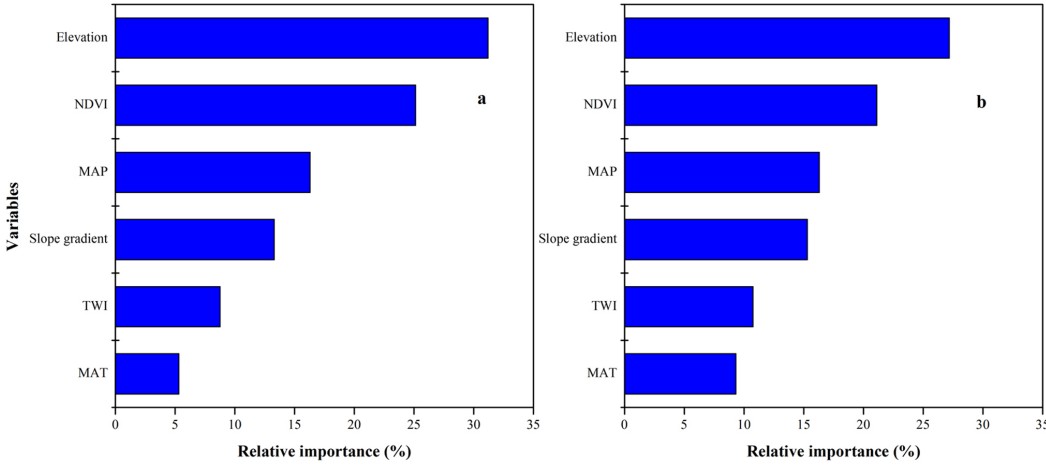

**Figure 8** Relative importance of each variable as determined from the boosted regression trees (BRT) model in predicting soil organic carbon (SOC) (A) and soil total nitrogen (STN) (B).

low-elevation areas, which were conducive to the accumulation of SOC and STN. Of the six climate-vegetation landscape units, soil under coastal plain had the lowest SOC and STN as also reported in *Causarano et al. (2008)*. The low SOC and STN in these landscape units might be caused by increased organic matter decomposition and erosion and tillage losses (*Burke et al., 1989; Lettens et al., 2004; Adhikari & Hartemink, 2015*).

We observed that with increasing elevation, this is true for forest pixels (Fig. 8). The effect of elevation on SOC and STN has been examined in a number of studies (e.g., *Kieft et al., 1998; Kunkel et al., 2011; Minasny et al., 2013; Adhikari et al., 2013; Chen et al., 2015; Minasny et al., 2017; Liang et al., 2019*). *Wang et al. (2017)* reported that SOC and STN significantly increased with elevation. Differences in elevation gradients may have affected the input and loss of soil carbon and nitrogen mainly through indirect factors along the

gradient, such as precipitation and temperature (*Minasny et al., 2013*; *Elbasiouny et al., 2014*; *Were et al., 2015*; *Adhikari & Hartemink, 2015*).

The predicted distribution of SOC and STN showed a discontinuous pattern (Figs. 7C and 7D). We found that the abrupt change in SOC and STN contents were located at the junction of landscapes. Such a pattern was due to the method of landscape division leading to an abrupt change of different landscape boundaries (*Jaeger, 2000*; *Peilin et al., 2010*). The apparent partitioning between landscapes led to spatially discontinuous distribution of SOC and STN, resulting in may lead to certain unreasonable predictions (*Rosenbloom et al., 2006*). This pattern may be attributed to the evident transition between the two landscapes, but the transition zone was smaller than the pixel resolution (*Belshe et al., 2012*).

### Limitation of the study

Although the ISA model constructed in this study had a good performance in predicting SOC and STN content, there were still some limitations. Firstly, we used a soil-landscape model to collect soil samples, which is idiosyncratically suitable for the ISA model. An independent probability sampling method would be required in our subsequent research. Secondly, this study area was divided into several typical climate-vegetation landscape units and then applied the model to simulate their SOC and STN. This treatment ignored the gradual transition between landscape units with respect to SOC and STN, which might have biased our regional prediction. In the future, the model should be revised to avoid this bias. Thirdly, because the study area was dominated by mountains, which were easily affected by the terrain and clouds, the high-altitude areas were prone to produce shadows in the process of image segmentation, resulting in large reflectivity errors of satellite image data. This might have also introduced prediction errors. Finally, this research was limited to topsoil (0–20 cm) SOC and STN content, which might have led to underestimating SOC and STN due to a large amount of SOC and STN typically stored in deeper soil layers in the region.

## CONCLUSIONS

This study advanced an ISA model to predict the spatial distribution of SOC and STN in the Northeast coastal area of China. The ISA model incorporated a GMMC method to divide the study area into six climatic-vegetation landscapes first. The six landscapes were then modeled using a t theory of environmental similarity. Comparing the prediction of GWR, RK, and BRT models, we found that the ISA model was the most robust and effective method in mapping SOC and STN, which explained 78% and 83% of the spatial variation of SOC and STN in the study area, respectively. Of the six climate-vegetation landscape units, the soils under low-elevation forest had the highest level of SOC and STN content than soils under the rest of the landscape units including coastal plain, interior plain, medium-coverage grasslands, low-mountain shrub, and high-coverage grasslands. Topography is the main driving force of SOC and STN distribution in the Northeast coastal area. Therefore, we suggested that terrain variables should be included in future SOC and TN mapping studies, especially in coastal and hilly areas of China. Overall, the improved ISA method better predicted SOC and STN in the region. Our predicted SOC

and STN distribution shall provide important information for soil and water conservation, ecological restoration, environmental management and agricultural production planning in the Northeast coastal agricultural region.

### Funding
This work was supported by the China Postdoctoral Science Foundation (2019M660782) and the young scientific and Technological Talents Project of Liaoning Province (Grant No. LSNQN201910 and LSNQN201914). The funders had no role in study design, data collection and analysis, decision to publish, or preparation of the manuscript.

### Grant Disclosures
The following grant information was disclosed by the authors:
China Postdoctoral Science Foundation: 2019M660782.
Young Scientific and Technological Talents Project of Liaoning Province: LSNQN201910, LSNQN201914.

### Competing Interests
Kabindra Adhikari is an Academic Editor for PeerJ.

### Author Contributions
- Shuai Wang conceived and designed the experiments, analyzed the data, prepared figures and/or tables, and approved the final draft.
- Kabindra Adhikari, Qianlai Zhuang, Qiubing Wang and Zhenxing Bian conceived and designed the experiments, authored or reviewed drafts of the paper, and approved the final draft.
- Zijiao Yang conceived and designed the experiments, performed the experiments, authored or reviewed drafts of the paper, and approved the final draft.
- Xinxin Jin conceived and designed the experiments, performed the experiments, analyzed the data, prepared figures and/or tables, and approved the final draft.

### Data Availability
The original data sampling point latitude and longitude and SOC and STN data values are available in the Supplementary Files.

### Supplemental Information
Supplemental information for this article can be found online at http://dx.doi.org/10.7717/peerj.9126#supplemental-information.

## REFERENCES

**Adhikari K, Hartemink AE. 2015.** Digital mapping of topsoil carbon content and changes in the Driftless Area of Wisconsin, USA. *Soil Science Society of America Journal* **79(1)**:155–164 DOI 10.2136/sssaj2014.09.0392.

# PeerJ

**Adhikari K, Hartemink AE, Minasny B, Kheir RB, Greve MB, Greve MH. 2014.** Digital mapping of soil organic carbon contents and stocks in Denmark. *PLOS ONE* **9(8)**:e105519 DOI 10.1371/journal.pone.0105519.

**Adhikari K, Kheir RB, Greve MB, Greve MH. 2013.** Comparing kriging and regression approaches for mapping soil clay content in a diverse Danish landscape. *Soil Science* **178(9)**:505–517 DOI 10.1097/SS.0000000000000013.

**Adhikari K, Owens PR, Ashworth AJ, Sauer TJ, Libohova Z, Richter JL, Miller DM. 2018.** Topographic controls on soil nutrient variations in a silvopasture system. *Agrosystems, Geosciences & Environment* **1(1)**:1–15.

**An YM, Yang L, Zhu AX, Qin CZ, Shi JJ. 2018.** Identification of representative samples from existing samples for digital soil mapping. *Geoderma* **311**:109–119 DOI 10.1016/j.geoderma.2017.03.014.

**Banfield JD, Banfield AE. 1993.** Model-based Gaussian and non-Gaussian clustering. *Biometrics* **49(3)**:803–821.

**Batjes NH. 1996.** Total carbon and nitrogen in the soils of the world. *European Journal of Soil Science* **47**:151–163 DOI 10.1111/j.1365-2389.1996.tb01386.x.

**Belshe EF, Schuur EAG, Bolker BM, Bracho R. 2012.** Incorporating spatial heterogeneity created by permafrost thaw into a landscape carbon estimate. *Journal of Geophysical Research: Biogeosciences* **117**:G01026 DOI 10.1029/2011JG001836.

**Bhunia GS, Shi PKumar, Pourghasemi HR. 2019.** Soil organic carbon mapping using remote sensing techniques and multivariate regression model. *Geocarto International* **34(2)**:215–226 DOI 10.1080/10106049.2017.1381179.

**Brunsdon C, Fotheringham S, Chariton M. 1998.** Geographically weighted regression-modeling spatial non-stationarity. *The Statistician* **47**:431–443.

**Burke IC, Yonker CM, Parton WJ, Cole CV, Schimel DS, Flach K. 1989.** Texture, climate, and cultivation effects on soil organic matter content in US grassland soils. *Soil Science Society of America Journal* **53(3)**:800–805 DOI 10.2136/sssaj1989.03615995005300030029x.

**Caminoserrano M, Gielen B, Luyssaert S, Ciais P, Vicca S, Guenet B, De Vos B, Cools N, Ahrens B, Altaf Arain M, Borken W, Clarke N, Clarkson B, Cummins T, Don A, Pannatier EG, Laudon H, Moore T, Nieminen TM, Nilsson MB, Peichl M, Schwendenmann L, Siemens J, Janssens IA. 2014.** Linking variability in soil solution dissolved organic carbon to climate, soil type, and vegetation type. *Global Biogeochemical Cycles* **28(5)**:497–509.

**Causarano HJ, Franzluebbers AJ, Shaw JN, Reeves DW, Raper RL, Wood C. 2008.** Soil organic carbon fractions and aggregation in the Southern Piedmont and Coastal Plain. *Soil Science Society of America Journal* **72(1)**:221–230 DOI 10.2136/sssaj2006.0274.

**Chen C, Hu K, Li H, Yun A, Li B. 2015.** Three-dimensional mapping of soil organic carbon by combining kriging method with profile depth function. *PLOS ONE* **10(6)**:e0129038 DOI 10.1371/journal.pone.0129038.

**Clement F, Orange D, Williams M, Mulley C, Epprecht M. 2009.** Drivers of afforestation in Northern Vietnam: assessing local variations using geographically weighted regression. *Applied Geography* **29**:561–576 DOI 10.1016/j.apgeog.2009.01.003.

**Conrad O, Bechtel B, Bock M, Dietrich H, Fischer E, Gerlitz L, Wehberg J, Wichmann V, Böhner J. 2015.** System for automated geoscientific analyses (SAGA) v.2.1.4. *Geoscientific Model Development* **8(7)**:1991–2007 DOI 10.5194/gmd-8-1991-2015.

**Danielsson PE. 1980.** Euclidean distance mapping. *Computer Graphics and Image Processing* **14(3)**:227–248 DOI 10.1016/0146-664X(80)90054-4.

**Davidson EA, Janssens IA. 2006.** Temperature sensitivity of soil carbon decomposition and feedbacks to climate change. *Nature* **440**:165–173 DOI 10.1038/nature04514.

**Dempster AP, Laird NM, Rubin DB. 1977.** Maximum Likelihood from Incomplete Data Via Em Algorithm. *Journal of the Royal Statistical Society* **39**:1–38.

**Ding J, Yang A, Wang J, Sagan V, Yu D. 2018.** Machine-learning-based quantitative estimation of soil organic carbon content by VIS/NIR spectroscopy. *PeerJ* **6**:e5714 DOI 10.7717/peerj.5714.

**Elbasiouny H, Abowaly M, Abu_Alkheir A, Gad A. 2014.** Spatial variation of soil carbon and nitrogen pools by using ordinary Kriging method in an area of north Nile Delta, Egypt. *Catena* **113**:70–78 DOI 10.1016/j.catena.2013.09.008.

**Farr TG, Kobrick M. 2000.** Shuttle Radar Topography Mission produces a wealth of data. *Eos, Transactions American Geophysical Union* **81(48)**:583–585 DOI 10.1029/EO081i048p00583.

**Fraley C, Raftery AE. 2002.** Model-based clustering, discriminant analysis, and density estimation. *Journal of the American Statistical Association* **97**:611–631 DOI 10.1198/016214502760047131.

**Friedman J, Hastie T, Tibshirani R. 2000.** Additive logistic regression: a statistical view of boosting. *The Annals of Statistics* **28(2)**:337–407.

**Fuchset H, Magdon P, Kleinn C, Flessa H. 2009.** Estimating aboveground carbon in a catchment of the Siberian forest tundra: Combining satellite imagery and field inventory. *Remote Sensing of Environment* **113(3)**:518–531 DOI 10.1016/j.rse.2008.07.017.

**GartenJr CT, Ashwood TL. 2002.** Landscape level differences in soil carbon and nitrogen: implications for soil carbon sequestration. *Global Biogeochemical Cycles* **16(4)**:61–1.

**Grimm R, Behrens T, Märker M, Elsenbeer H. 2008.** Soil organic carbon concentrations and stocks on Barro Colorado Island—Digital soil mapping using Random Forests analysis. *Geoderma* **146(1)**:102–113 DOI 10.1016/j.geoderma.2008.05.008.

**Hengl T, Heuvelink G, Stein A. 2004.** A Generic Framework for Spatial Prediction of Soil Variables Based on Regression Kriging. *Geoderma* **122(1-2)**:75–93.

**Huang H, Cheng X, Li B, Liu Y. 2007.** Application of High-Resolution Remote Sensing Image Using ENVI. *Geospatial Information* **3**:41–43.

**Hudson BD. 1992.** The soil survey as paradigm-based science. *Soil Science Society of America Journal* **56**:836–841 DOI 10.2136/sssaj1992.03615995005600030027x.

**Jaeger JAG. 2000.** Landscape division, splitting index, and effective mesh size: new measures of landscape fragmentation. *Landscape Ecology* **15(2)**:115–130 DOI 10.1023/A:1008129329289.

Jenny H. 1941. *Factors of Soil Formation.* New York: McGraw-Hill.

Jobbagy EG, Jackson RB. 2000. The vertical distribution of soil organic carbon and its relation to climate and vegetation. *Ecological Applications* **10**:423–436 DOI 10.1890/1051-0761(2000)010[0423:TVDOSO]2.0.CO;2.

Kieft TL, White CS, Loftin SR, Aguilar R, Craig J, Skaar DA. 1998. Temporal dynamics in soil carbon and nitrogen resources at a grassland–shrubland ecotone. *Ecology* **79(2)**:671–683.

Kovačević M, Bajat B, Gajić B. 2010. Soil type classification and estimation of soil properties using support vector machines. *Geoderma* **154(3)**:340–347 DOI 10.1016/j.geoderma.2009.11.005.

Kumar S, Lal R, Liu D. 2012. A geographically weighted regression kriging approach for mapping soil organic carbon stock. *Geoderma* **189**:627–634.

Kunkel ML, Flores AN, Smith TJ, McNamara JP, Benner SG. 2011. A simplified approach for estimating soil carbon and nitrogen stocks in semi-arid complex terrain. *Geoderma* **165(1)**:1–11 DOI 10.1016/j.geoderma.2011.06.011.

Lal R. 2004. Soil carbon sequestration impacts on global climate change and food security. *Science* **304(5677)**:1623–1627 DOI 10.1126/science.1097396.

Lettens S, Van Orshoven J, Van Wesemael B, Muys B. 2004. Soil organic and inorganic carbon contents of landscape units in Belgium derived using data from 1950 to 1970. *Soil Use and Management* **20(1)**:40–47 DOI 10.1079/SUM2003221.

Liang Z, Chen S, Yang Y, Zhou Y, Shi Z. 2019. High-resolution three-dimensional mapping of soil organic carbon in China: Effects of SoilGrids products on national modeling. *Science of The Total Environment* **685**:480–489 DOI 10.1016/j.scitotenv.2019.05.332.

Liu F, Rossiter D-G, Song XD, Zhang GL, Yang RM, Zhao YG, Ju B. 2016. A similarity-based method for three-dimensional prediction of soil organic matter concentration. *Geoderma* **263**:254–263 DOI 10.1016/j.geoderma.2015.05.013.

Liu J. 2010. *Mapping Soil Properties Using Individual Representativeness of Samples over Large Area.* Beijing: Beijing Normal University (in Chinese).

Matejovic I. 1993. Determination of carbon, hydrogen, and nitrogen in soils by automated elemental analysis (dry combustion method). *Communications in Soil Science and Plant Analysis* **24(17-18)**:2213–2222 DOI 10.1080/00103629309368950.

McBratney AB, Santos MLM, Minasny B. 2003. On digital soil mapping. *Geoderma* **17**:3–52.

Minasny B, Malone BP, McBratney AB, Angers DA, Arrouays D, Chambers A, Field DJ. 2017. Soil carbon 4 per mille. *Geoderma* **292**:59–86 DOI 10.1016/j.geoderma.2017.01.002.

Minasny B, McBratney AB. 2008. Regression rules as a tool for predicting soil properties from infrared reflectance spectroscopy. *Chemometrics and Intelligent Laboratory Systems* **94(1)**:72–79 DOI 10.1016/j.chemolab.2008.06.003.

Minasny B, McBratney AB, Malone BP, Wheeler I. 2013. Digital mapping of soil carbon. *Advances in Agronomy* **118**:1–47 DOI 10.1016/B978-0-12-405942-9.00001-3.

**Mondal A, Khare D, Kundu S, Mondal S, Mukherjee S, Mukhopadhyay A. 2017.** Spatial soil organic carbon (SOC) prediction by regression kriging using remote sensing data. *The Egyptian Journal of Remote Sensing and Space Science* **20**(**1**):61–70 DOI 10.1016/j.ejrs.2016.06.004.

**Moore ID, Gessler PE, Nielsen GAE, Peterson GA. 1993.** Soil attribute prediction using terrain analysis. *Soil Science Society of America Journal* **57**(**2**):443–452 DOI 10.2136/sssaj1993.03615995005700020026x.

**Mulder VL, Lacoste M, Richer-de Forges AC. 2016.** National versus global modelling the 3D distribution of soil organic carbon in mainland France. *Geoderma* **263**:16–34 DOI 10.1016/j.geoderma.2015.08.035.

**Odeh IOA, McBratney AB, Chittleborough DJ. 1995.** Further results on prediction of soil properties from terrain attributes: heterotopic cokriging and regression-kriging. *Geoderma* **67**:215–226 DOI 10.1016/0016-7061(95)00007-B.

**Peilin L, Chunla L, Yunyuan D, Xiuying SHEN, Bohua LI, Zui HU. 2010.** Landscape division of traditional settlement and effect elements of landscape gene in China. *Acta Geographica Sinica* **65**(**12**):1496–1506.

**Post WM, Kwon KC. 2000.** Soil carbon sequestration and land-use change: processes and potential. *Global Change Biology* **6**:317–327 DOI 10.1046/j.1365-2486.2000.00308.x.

**R Development Core Team. 2013.** R: a language and environment for statistical computing. Vienna: R Foundation for Statistical Computing.

**Rial M, Cortizas AM, Taboada T, Rodríguez-Lado L. 2017.** Soil organic carbon stocks in Santa Cruz Island, Galapagos, under different climate change scenarios. *Catena* **156**:74–81 DOI 10.1016/j.catena.2017.03.020.

**Rosenbloom NA, Harden JW, Neff JC, Schimel DS. 2006.** Geomorphic control of landscape carbon accumulation. *Journal of Geophysical Research: Biogeosciences* **111**:G01004 DOI 10.1029/2005JG000077.

**Schad P, Van Huyssteen C, Micheli E. 2014.** *International soil classification system for naming soils and creating legends for soil maps; world reference base for soil resources.* Vol. 106. Rome, Italy: WRB, 246–385.

**Scull P, Franklin J, Chadwick OA, McArthur D. 2003.** Predictive soil mapping: a review. *Progress in Physical Geography* **27**(**2**):171–197 DOI 10.1191/0309133303pp366ra.

**Shi X, Zhu AX, Burt JE, Qi F, Simonson D. 2004.** A case-based reasoning approach to fuzzy soil mapping. *Soil Science Society of America Journal* **68**(**3**):885–894 DOI 10.2136/sssaj2004.8850.

**Sollins P, Homann PS, Caldwell BA. 1996.** Stabilization and destabilization of soil organic matter: mechanisms and controls. *Geoderma* **74**:65–105.

**Song XD, Brus DJ, Liu F, Li DC, Zhao YG, Yang JL, Zhang GL. 2016.** Mapping soil organic carbon content by geographically weighted regression: a case study in the Heihe River Basin, China. *Geoderma* **261**:11–22 DOI 10.1016/j.geoderma.2015.06.024.

**Stevens A, Udelhoven T, Denis A, Tychon B, Lioy R, Hoffmann L, Van Wesemael B. 2010.** Measuring soil organic carbon in croplands at regional scale using airborne imaging spectroscopy. *Geoderma* **158**(**1**):32–45 DOI 10.1016/j.geoderma.2009.11.032.

**Taghizadeh-Mehrjardi R, Minasny B, Sarmadian F, Malone BP. 2014.** Digital mapping of soil salinity in Ardakan region, central Iran. *Geoderma* **213**:15–28 DOI 10.1016/j.geoderma.2013.07.020.

**Tesfaye MA, Bravo F, Ruiz-Peinado R, Pando V, Bravo-Oviedo A. 2016.** Impact of changes in land use, species and elevation on soil organic carbon and total nitrogen in Ethiopian Central Highlands. *Geoderma* **261**:70–79 DOI 10.1016/j.geoderma.2015.06.022.

**Wang K, Zhang C, Li W. 2013.** Predictive mapping of soil total nitrogen at a regional scale: a comparison between geographically weighted regression and cokriging. *Applied Geography* **42**:73–85 DOI 10.1016/j.apgeog.2013.04.002.

**Wang S, Adhikari K, Wang Q, Jin X, Li H. 2018a.** Role of environmental variables in the spatial distribution of soil carbon (C), nitrogen (N), and C: N ratio from the northeastern coastal agroecosystems in China. *Ecological Indicators* **84**:263–272 DOI 10.1016/j.ecolind.2017.08.046.

**Wang S, Wang Q, Adhikari K, Jia S, Jin X, Liu H. 2016.** Spatial-temporal changes of soil organic carbon content in Wafangdian, China. *Sustainability* **8(11)**:1154 DOI 10.3390/su8111154.

**Wang S, Zhuang Q, Jia S, Jin X, Wang Q. 2018b.** Spatial variations of soil organic carbon stocks in a coastal hilly area of China. *Geoderma* **314**:8–19 DOI 10.1016/j.geoderma.2017.10.052.

**Wang S, Zhuang Q, Wang Q, Jin X, Han C. 2017.** Mapping stocks of soil organic carbon and soil total nitrogen in Liaoning Province of China. *Geoderma* **305**:250–263 DOI 10.1016/j.geoderma.2017.05.048.

**Were K, Bui DT, Dick ØB, Singh BR. 2015.** A comparative assessment of support vector regression, artificial neural networks, and random forests for predicting and mapping soil organic carbon stocks across an Afromontane landscape. *Ecological Indicators* **52**:394–403 DOI 10.1016/j.ecolind.2014.12.028.

**Xu Y, Smith SE, Grunwald S, Abd-Elrahman A, Wani SP, Nair VD. 2018.** Estimating soil total nitrogen in smallholder farm settings using remote sensing spectral indices and regression kriging. *Catena* **163**:111–122 DOI 10.1016/j.catena.2017.12.011.

**Yang L, Huang C, Liu G, Liu J, Zhu A-X. 2015a.** Mapping soil salinity using a similarity-based prediction approach: a case study in Huanghe River Delta, China. *Chinese Geographical Science* **25(3)**:283–294 DOI 10.1007/s11769-015-0740-7.

**Yang L, Qi F, Zhu AX, Shi JJ, An YM. 2016a.** Evaluation of Integrative Hierarchical Stepwise Sampling for Digital Soil Mapping. *Soil Science Society of America Journal* **80(3)**:637–651 DOI 10.2136/sssaj2015.08.0285.

**Yang RM, Rossiter DG, Liu F, Lu Y, Yang F, Yang F. 2015b.** Predictive mapping of topsoil organic carbon in an Alpine environment aided by Landsat TM. *PLOS ONE* **10(10)**:e0139042 DOI 10.1371/journal.pone.0139042.

**Yang L, Zhu AX, Qi F, Qin CZ, Li BL, Pei T. 2013.** An integrative hierarchical stepwise sampling strategy and its application in digital soil mapping. *International Journal of Geographical Information Science* **27(1)**:1–23 DOI 10.1080/13658816.2012.658053.

**Yang RM, Zhang GL, Liu F, Lu YY, Yang F, Yang F, Li DC. 2016b.** Comparison of boosted regression tree and random forest models for mapping topsoil organic carbon concentration in an alpine ecosystem. *Ecological Indicators* **60**:870–878 DOI 10.1016/j.ecolind.2015.08.036.

**Zhao MS, Zhang GL, Wu YJ, Li DC, Zhao Y-G. 2015.** Driving forces of soil organic matter change in Jiangsu Province of China. *Soil Use and Management* **31(4)**:440–449 DOI 10.1111/sum.12206.

**Zhu AX. 1997.** A similarity model for representing soil spatial information. *Geoderma* **77**:217–242 DOI 10.1016/S0016-7061(97)00023-2.

**Zhu AX, Liu J, Qin CZ, Zhang SJ, Chen YN, Ma XW. 2010.** Soil property mapping over large areas using sparse ad -hoc samples. In: *19th World Congress of Soil Science, Soil Solutions for a Changing World*. Brisbane, Australia: The International Union of Soil Science, (accessed on 1-6 August 2010).

**Zhu AX, Yang L, Li B, Qin C, English E, Burt JE, Zhou C. 2008.** Purposive sampling for digital soil mapping for areas with limited data. In: Hartemink AE, ed. *Digital soil mapping with limited data*. Berlin: Springer, 33–245.