# Peer review of "An improved similarity-based approach to predicting and mapping soil organic carbon and soil total nitrogen in a coastal region of northeastern China"

_PeerJ, doi:10.7717/peerj.9126_

## Round 0.1 · original submission · Major Revisions

Attached please find the reviewers comments on your manuscript. Revisions will need to be made for reconsideration for publication.

Reviewer 1 ·

Basic reporting

The paper evaluates several methods for mapping SOC and TN in an area in China. The paper is mostly well written. Methods are well-described. However there are some statements that cannot be generalised and not true. Those sentences need to be revised.
For example
L. 57: The classification is not meaningful and not true: decision tree is part of machine learning models.
L. 64 The statement i snot true. Geostatistics can be applied at any scale
L. 147. The apprach grouped the landscape into clusters and then samples were taken from each cluster. This is a stratified sampling, not purposive sampling.
Eq. 8 is called the confusion index or consistency of prediction. It has nothing to do with the model uncertainty.


There are also few English expression that can be improved, e.g.
L. 379 Less steady??
Table 1: "Description" not "describe"

Experimental design

The experiment is well conducted.
- Sampling
L. 142, It should start with the description of the area, covariates and sampling design. Please describe GMCC method
The data should be described later.
Eq. 7 What is m? How many m? How was m determined?
Why not use Euclidean distance standardised by std. deviation of data?

Validity of the findings

The finding is applicable for the location in the area. I am surprised that the similarity method performed much better than regression methods. Since regression methods are calibrated from the data while the similarity method has to assume that the classes are representative of the data. Does the clustering done after the sample was collected?
How was the actual sample determined? Maybe the sample was biased and not based on probability sampling. This needs to be discussed.

Additional comments

The paper is generally ok, but a major revision is needed to soften the statements and define limitation of the studey.

Reviewer 2 ·

Basic reporting

I think the manuscript writing should be greatly improved. The manuscript is superficial or lacking information in some parts, especially the introduction and discussion. I suggest that English editing and proofreading by a native speaker.
Pay attention and avoid unnecessary repetitions.

INTRO
It would be important to briefly clarify how the auxiliary variables: elevation, climate (temperatures) and landscape units determine the C and N cycles (and variations), especially on a local scale.

R&D
I recommend comparing the results obtained (SOC - STN) with more previous research. Notably, because there were many previous studies on mapping & modeling SOC - STN in coastal ecosystems.

Figs.
Improve the quality of the figures.

Experimental design

The research project is within the aims and scope of the journal. The technique and methodology are reasonable, and the sample size is appropriate for the purpose of the project.
The results are sufficiently valuable and the methodological approach could be of interest to some future readers.

I suggest reading:
Spatial soil organic carbon (SOC) prediction by regression kriging using remote sensing data (Mondal et al., 2017) to improve RK paragraph.

Only few questions:
Why did you choose these auxiliary variables? And which conversion process has been used?

The NDVI calculation procedures have been drafted in an excessively concise manner.
Were Landsat atmospheric correction algorithms implemented? Average annual NDVI?

Validity of the findings

Data is sufficiently robust, statistically sound and cross validation controlled.

The conclusions are linked to original research question.

Additional comments

I believe it is appropriate to add the SOC and STN distribution maps drawn up through GWR & RK, at least in the supplementary materials.

Figs.
F.1 I recommend changing the color of the sampling points, and adding the coordinate grid labels.
F.5 Include the units of measurement in the two graphs.
F.6 The legends are not homogeneous, I recommend adapting their dimensions.
F.6c g kg-1 apex!

Reviewer 3 ·

Basic reporting

This is a clear research paper with an acceptable storyline, it is well structured and discussed. There is sufficient information in the introduction.

The first table [line 164]was Table 5, Table 1 should be converted to a matrix of maps as one figure in the Field sample data [line 143]

[line 187:199] might be suitable for introduction, no need to talk about soil formation in the Model Development section.

Figure 3 has to be in the study area description, too weak finding to be in the results and discussion section. No need for Figure 4, table 1 has almost the same information!

Experimental design

This topic is well fit to the PeerJ aims and scopes. The research topic is very exciting and the objectives are promising.

The research questions are well defined, relevant and meaningful.

This is quite old fashion research, nothing new, the methodology is not that important nor the data, using just NDVI delivered from Landsat 5 which was March 1984.

It is clear that the authors didn't make the efforts to address these questions using adequate data and tools.

Example of a similar research topic and case study but with higher quality:

Zongzheng Liang, Songchao Chen, Yuanyuan Yang, Yue Zhou, Zhou Shi,
High-resolution three-dimensional mapping of soil organic carbon in China: Effects of SoilGrids products on national modeling, Science of The Total Environment, Volume 685, 2019,

Validity of the findings

The prediction modeling approach needs to be higher complex in order to map soil organic carbon and soil total nitrogen. The similarity-based approach, geographically weighted regression, and kriging are not enough to build this trust in your maps' reliability.

For example, Figure 7. You can't have a vertical prediction to SOC and STN, just impossible!!

The conclusion section is a reflection of how weak this research work.

Additional comments

This research is well written and structured, unfortunately, it needs a better scientific approach, data, and modeling.

---

## Round 0.2 · Minor Revisions

The reviewers are overall satisfied with the corrections made. Please correct all minor revisions and resubmit.

Reviewer 1 ·

Basic reporting

The paper has been improved. Nevertheless there is still few awkward expression that would benefit from another round of edits.
e.g. (one of many)
L. 26 would help
L. 32 Soil samples from the depth of 0-20 cm were
L. 39 How much improvement compared to other methods?
L 220 there is only a matrix

Fig 7 a & b. please clarify what does the consistency mean,what do high values means?

etc.. Please go through the manuscript carefully

Experimental design

The experimental design is well described.
It has been revised as sugggested

Validity of the findings

The authors need to add limitation of the study. As the samples were collected based on an assumed soil-landscape model, it may favour the ISA model.
Ideally, an independent probability sampling would be required. Limitations should be discussed.

L. 428-430, please do not compare R2 values with other studies, it is not valid. R2 values depend on the variance of the data.

Reviewer 2 ·

Basic reporting

M Clean

The English editing has been significantly improved according with proposed corrections and suggestions, however, some imperfections persist.
Examples:
47 – 50 “And” was used four times in two lines of the same sentence (improve the editing).
57 Landsat TM imagery aren't environmental factors but tools to describe the spatial patterns.
86 Indicate soil properties
180 coordinates
181-186 The paragraph is confusing

Experimental design

OK

MM
also specify that the samples were collected in 2016 as previously done in the abstract

Validity of the findings

OK

Additional comments

OK

The Figs have been modified and improved

---

## Round 0.3 · accepted · Accept

Thank you for the revisions made. The ms is now accepted for publication.